# OpenReview forum: "Energy-Based Reward Models for Robust Language Model Alignment"
_colmweb.org/COLM/2025/Conference — COLM 2025_

### Official Review · Reviewer_7x94 · 2025-05-10

**Rating:** 6
**Confidence:** 4
**Ethics Flag:** 1

**Summary:**

Very interesting and potentially very practical approach for improving quality of trained RMs.
The method has 2 downsides: (1) seem to require access to RM's training data and (2) hyperparameter choices for noise "we perturb the rewards with noise νi ∼" can be tricky.
Having said that for people training their own RMs this is still potentially very useful.

The main downside is the choice of baseline RM - something with a score of 55 on RewardBench, whereas RewardBench has RMs with open training data and of same size or smaller with average scores over 90.

**Questions To Authors:**

Please chose the stronger RM as baseline, I'll be happy to increase my score.

**Reasons To Accept:**

Very interesting method applicable on top of existing RMs. Basically it gives people who train their own RMs addition tool to make their RM and RLHF pipelines more robust. In this respect the proposed method in very appealing.

I also like that authors not only reported benchmarks related solely to RM training but also reported RLHF results.

**Reasons To Reject:**

Very sad choice of baseline RM :(
I don't understand why the authors pick as base RM 70B Pythia-based RM that gets 55.10 average on RewardBench whereas there are tens of RMs, including 8B ones, and models RMs with open RM training data which have all scores above 90.0 average.

---

> ### Author Response · Authors · 2025-06-03
> **Response to  Reviewer 7x94 (1/2)**
>
> Thank you for your thoughtful and constructive feedback. We're glad you found the paper interesting.
>
> **1. Seem to require access to RM's training data**
>
> EBRM does require access to the Base RM’s training data. This is a **natural consequence of its design as a post-hoc refinement method**. Training on the same preference data avoids distribution shift, ensures compatibility with the RM’s internal representations, and enables EBRM to refine the reward function rather than relearn it from scratch. We view removing this requirement as an interesting direction for future work.
>
> **2. Hyperparameter choice for noise can be tricky**
>
> We agree that tuning the noise hyperparameter can be tricky. Injecting noise into reward targets allows the model to better adapt across tasks with varying ambiguity. As shown in Section E, Figure 5, extreme values of $\beta$ (e.g., 0.01 or 0.5) reduce performance gain, but a moderate setting (0.05-0.2) yields consistent overall gains. This indicates that EBRM is **robust to a wide range of reasonable settings**. In practice, a simple validation sweep on tasks of differing ambiguity is sufficient to select a robust $\beta$ that generalizes well.
>
> **3. Clarification on the Base RM Choice**
>
> We would like to clarify a key misunderstanding: the Base RM used in our initial experiments was **70M**-parameter
> Pythia-based model - not 70B. Smaller RMs like the 70M variant are more susceptible to overconfidence, miscalibration, and reward hacking, making them particularly suitable for evaluating robustness and the effectiveness of post-hoc refinement methods such as EBRM.
>
> Further, comparing against ensemble baselines (3× RMs) is computationally feasible only at small scales. Using a small Base RM also allows us to leverage stronger reward models (AlpacaFarm 7B Human Preference Reward Model) as "gold" RMs for RL experimentation.

---

> > ### Author Response · Authors · 2025-06-03
> > **Response to Reviewer 7x94 (2/2)**
> >
> > **4. Scalability**
> >
> > We agree that benchmarking against stronger RMs is important and have addressed this by extending our experiments to include larger and competitive models.
> >
> > In particular, we now evaluate on **2.6B Pythia Reward Model** and **Skywork-LLaMA3.1-8B** RM [2]. The Skywork RM is ranked \#11 on the RewardBench leaderboard and the strongest publicly available 8B reward model. Results are summarized below:
> >
> > Win rates on RewardBench for the 2.6B Pythia Reward Model.
> > |         | Chat      | Chat-Hard | Safety    | Reasoning | Average   |
> > |---------|-----------|-----------|-----------|-----------|-----------|
> > | Base RM | **81.01** | 35.20     | 38.26     | 73.60     | 57.02     |
> > | EBRM    | 79.61     | **36.73** | **42.69** | **73.84** | **58.22** |
> >
> > Win rates on Reward Model Benchmark (RMB) for the 2.6B Pythia Reward Model.
> > | Method  |           | Harmlessnes |           | Helpfulness | Average   |
> > |---------|-----------|-------------|-----------|-------------|-----------|
> > |         | Pairwise  | BoN         | Pairwise  | BoN         |           |
> > | Base RM | 51.67     | 36.10       | 45.08     | 27.65       | 40.13     |
> > | EBRM    | **53.45** | **36.95**   | **48.89** | **31.23**   | **42.63** |
> >
> > Win rates on RewardBench for the 8B Skywork Reward Model.
> > | Method  | Chat      | Chat-Hard | Safety    | Reasoning | Average   |
> > |---------|-----------|-----------|-----------|-----------|-----------|
> > | Base RM | **94.69** | 88.60     | 92.58     | **96.70** | 93.14     |
> > | EBRM    | **94.69** | **89.69** | **92.66** | **96.70** | **93.43** |
> >
> > Win rates on Reward Model Benchmark (RMB) for the 8B Skywork Reward Model.
> > | Method  |           | Harmlessnes |           | Helpfulness | Average   |
> > |---------|-----------|-------------|-----------|-------------|-----------|
> > |         | Pairwise  | BoN         | Pairwise  | BoN         |           |
> > | Base RM | 72.39     | **56.37**   | 76.36     | 61.64       | 66.69     |
> > | EBRM    | **72.43** | 56.26       | **77.01** | **61.75**   | **66.86** |
> >
> > We observe a consistent positive gain in performance as the Base RM size increases. On the 2.6B Pythia-based RM, EBRM improves average scores on both RMB (+2.5\%) and RewardBench (+1.2\%) tasks. EBRM continues to provide measurable improvement even at the high end of the performance spectrum, delivering +0.29\% on RewardBench and +0.17\% on RMB over the already strong Skywork RM.
> >
> > For comparison, the recent uncertainty-aware method (URM) [1] required full RM retraining, shifting from Bradley-Terry to multi-attribute regression loss and achieved a +0.4\% gain on RewardBench. EBRM, by contrast, is plug-and-play: no retraining, no architecture changes, and still comparable gains.
> > We also note that our Skywork evaluation uses the **cleaned data version**, unlike prior work that unknowingly used a version with RewardBench data leakage [2]. Due to the lack of open-source code, we are unable to reproduce a direct comparison with URM. We also omit ensemble baselines here due to compute constraints.
> >
> > **In summary:** Our new results show that EBRM adds value even to top-tier RMs, reinforcing its relevance for practical and scalable alignment refinement.
> >
> > [1] Uncertainty-aware Reward Model: Teaching Reward Models to Know What is Unknown, arXiv preprint, 2024
> >
> > [2] Skywork-Reward: Bag of Tricks for Reward Modeling in LLMs, CoRR, 2024

---

> > > ### Author Response · Authors · 2025-06-08
> > > **Gentle Reminder**
> > >
> > > Dear Reviewer,
> > >
> > > Thank you once again for your valuable feedback and questions. We have provided detailed responses and additional empirical results, and we hope these address your concerns.
> > >
> > > As the discussion period concludes, we kindly encourage you to review our responses and **let us know if you have any remaining questions or concerns**.
> > >
> > > We sincerely appreciate your time and effort in helping us improve our work.
> > >
> > > Sincerely,
> > >
> > > Authors

---

> > > ### Comment · Reviewer_7x94 · 2025-06-08
> > >
> > > Thank you for extending your experiments. I've increased my score 5->6

---

### Official Review · Reviewer_gRuK · 2025-05-17

**Rating:** 7
**Confidence:** 4
**Ethics Flag:** 1

**Summary:**

This paper introduces a new framework for reward modeling (RM) that is based on a conditional energy-based model trained on top of a regular reward model, which enhances the robustness and generalization. Unlike standard RMs, it is able to capture uncertainty in the annotations and mitigate the impact of errors. The framework includes several techniques to achieve this, namely filtering out misaligned pairs from training data, using a label-noise-aware contrastive learning objective and initializing the model in hybrid way favoring reasonable RM scores within a predefined range. Empirical evaluation on common alignment  benchmarks shows improvements of up to 5.97% in safety-critical tasks.

**Questions To Authors:**

1. The paper demonstrates great results with EBRM as a post-hoc refinement approach. Have you considered incorporating energy-based principles directly during initial reward model training rather than as a separate step?
2. Your results show different performance patterns when scaling from 44M to 1.3B models, with smaller overall gains on the larger model. Do you expect this trend to continue with even larger models, and have you identified specific factors that might affect scaling behavior?
3. The increased inference time (~2x slower) could be problematic for production deployment. Have you explored optimization techniques to reduce this overhead while maintaining performance benefits?
4. How does EBRM compare empirically with uncertainty-aware reward models like those from Lou et al. and Yan et al.? While you mention they require more computational overhead, are there scenarios where their probabilistic heads might outperform EBRM's energy-based refinement?
5. Have you considered modifying the PPO or other RL algorithm to directly leverage this uncertainty information from EBRM?

**Reasons To Accept:**

- This paper explores an interesting application of energy-based models to a lightweight post-hoc refinement of reward models. It's well motivated and addresses important limitations in the standard reward modeling procedures.
- Achieves consistent improvements up to 5.97% with minimal memory overhead compared to standard RM and ensemble methods in RM benchmarks, performing especially well on tasks that emphasize safety.
- The method's compatibility with any existing reward model architecture makes it highly practical and likely to see broad adoption within the alignment research community.
- Improves performance in reinforcement learning scenarios with proxy-policy optimization compared to a standard RM.

**Reasons To Reject:**

- Increased inference time (~2x slower than base RM) may be problematic for real-time applications and use in production settings.
- While the approach shows strong results on top of a standard RM, additional evaluation on a more diverse set of RMs would strengthen generalizability claims.
- Comparisons are limited primarily to ensemble methods rather than a more diverse set of reward modeling methods.
- When scaling to larger models (1.3B), performance improvements show different patterns across tasks with smaller overall gains, raising questions about how consistently the benefits generalize across model scales.

---

> ### Author Response · Authors · 2025-06-03
> **Response to Reviewer gRuK (1/2)**
>
> Thank you for the detailed and insightful review. We are glad that you found our approach well-motivated, practical, and promising for broader adoption.
>
> **1. Inference Time**
>
> We believe EBRM offers a compelling cost-benefit trade-off, especially compared to alternatives like ensembling and retraining.
>
> i. EBRM increases inference latency by only $\sim$ 2× compared to the Base RM. In contrast, ensembles typically incur $\sim$n× the inference time and parameter count, where n $\ge$ 3 in practice. Retraining full RMs is substantially more expensive in both compute and time. EBRM can be trained in just 465 seconds from a 70M Pythia model with only 3\% the size of a standard RM.
>
> ii. Across five alignment tasks, EBRM outperforms ensembles by 1–5\% while being 34\% faster and adding $<$3\% in parameter count. These gains are non-trivial given that ensembles are a strong and widely adopted baseline.
>
> iii. EBRM requires no retraining and is applied post hoc to any RM, making it a lightweight and practical solution for deployment scenarios where retraining or maintaining ensembles is infeasible.
>
> **2. Scaling Performance**
>
> We agree that benchmarking against stronger RMs is important and have addressed this by extending our experiments to include larger and competitive models.
>
> In particular, we now evaluate on **2.6B Pythia Reward Model** and **Skywork-LLaMA3.1-8B** RM [2]. The Skywork RM is ranked \#11 on the RewardBench leaderboard and the strongest publicly available 8B reward model. Results are summarized below:
>
> Win rates on RewardBench for the 2.6B Pythia Reward Model.
> |         | Chat      | Chat-Hard | Safety    | Reasoning | Average   |
> |---------|-----------|-----------|-----------|-----------|-----------|
> | Base RM | **81.01** | 35.20     | 38.26     | 73.60     | 57.02     |
> | EBRM    | 79.61     | **36.73** | **42.69** | **73.84** | **58.22** |
>
> Win rates on Reward Model Benchmark (RMB) for the 2.6B Pythia Reward Model.
> | Method  |           | Harmlessnes |           | Helpfulness | Average   |
> |---------|-----------|-------------|-----------|-------------|-----------|
> |         | Pairwise  | BoN         | Pairwise  | BoN         |           |
> | Base RM | 51.67     | 36.10       | 45.08     | 27.65       | 40.13     |
> | EBRM    | **53.45** | **36.95**   | **48.89** | **31.23**   | **42.63** |
>
> Win rates on RewardBench for the 8B Skywork Reward Model.
> | Method  | Chat      | Chat-Hard | Safety    | Reasoning | Average   |
> |---------|-----------|-----------|-----------|-----------|-----------|
> | Base RM | **94.69** | 88.60     | 92.58     | **96.70** | 93.14     |
> | EBRM    | **94.69** | **89.69** | **92.66** | **96.70** | **93.43** |
>
> Win rates on Reward Model Benchmark (RMB) for the 8B Skywork Reward Model.
> | Method  |           | Harmlessnes |           | Helpfulness | Average   |
> |---------|-----------|-------------|-----------|-------------|-----------|
> |         | Pairwise  | BoN         | Pairwise  | BoN         |           |
> | Base RM | 72.39     | **56.37**   | 76.36     | 61.64       | 66.69     |
> | EBRM    | **72.43** | 56.26       | **77.01** | **61.75**   | **66.86** |
>
> We observe a consistent positive gain in performance as the Base RM size increases. On the 2.6B Pythia-based RM, EBRM improves average scores on both RMB (+2.5\%) and RewardBench (+1.2\%) tasks. EBRM continues to provide measurable improvement even at the high end of the performance spectrum, delivering +0.29\% on RewardBench and +0.17\% on RMB over the already strong Skywork RM.
>
> For comparison, the recent uncertainty-aware method (URM) [1] required full RM retraining, shifting from Bradley-Terry to multi-attribute regression loss and achieved a +0.4\% gain on RewardBench. EBRM, by contrast, is plug-and-play: no retraining, no architecture changes, and still comparable gains.
> We also note that our Skywork evaluation uses the **cleaned data version**, unlike prior work that unknowingly used a version with RewardBench data leakage [2]. Due to the lack of open-source code, we are unable to reproduce a direct comparison with URM. We also omit ensemble baselines here due to compute constraints.
>
> **In summary,** our new results show that EBRM adds value even to top-tier RMs, reinforcing its relevance for practical and scalable alignment refinement.

---

> > ### Author Response · Authors · 2025-06-03
> > **Response to Reviewer gRuK (2/2)**
> >
> > **3. When scaling to larger models (1.3B), performance improvements show different patterns across tasks with smaller overall gains**
> >
> > The table below summarises the absolute win-rate gains EBRM delivers over four Base RM sizes.
> > | Base RM        | Δ RewardBench ↑ | Δ RMB ↑   |
> > | -------------- | --------------- | --------- |
> > | Pythia 70M     | +1.06 pts       | +3.32 pts |
> > | Pythia 1.3B    | +0.95 pts       | +2.35 pts |
> > | Pythia 2.6B    | +1.20 pts       | +2.50 pts |
> > | Skywork 8B     | +0.29 pts       | +0.17 pts |
> >
> > While per-task gains vary slightly with scale due to differences in task ambiguity, difficulty, and the Base RM, the overall improvements remain stable and strictly positive across all model sizes.
> >
> > Importantly, even with the Skywork 8B RM [2], which ranks among the top models on RewardBench, EBRM still delivers measurable gains compared to other works (Response 2). This confirms that our method scales effectively and adds value even when applied to strong, highly trained, near-saturated base reward models.
> >
> > In summary, the task-level variance is expected, but the consistency of overall gains demonstrates that EBRM generalizes well across model scales and task types.
> >
> > **4. Baseline Comparison**
> >
> > Our baseline choices are grounded in practical constraints and design considerations.
> >
> > - Existing uncertainty-aware RMs do not have public implementations, which prevents direct comparison. Moreover, these methods typically require full retraining of the reward model, which is orthogonal to EBRM’s post-hoc and lightweight design.
> >
> > - Instead, we benchmark against ensembles, which are widely recognized for improving RM's robustness. While ensemble-based approaches are not
> > strictly post-hoc as they require multiple trained reward models, they represent a similar strategy for refining reward signals without modifying the core RM architecture.
> >
> > - Additionally, we include an indirect comparison to an existing uncertainty-aware method [1] via the Skywork-8B RM [2] in Response 2.  EBRM achieves comparable improvements on this strong baseline (+0.29\% on RewardBench, +0.17\% on RMB), demonstrating its effectiveness even without any RM retraining.
> >
> >
> > **5. Energy Modeling During RM Training or RL Algorithms**
> > Our primary goal was to develop a lightweight, post-hoc refinement method that can be
> > applied to existing reward models without retraining. However, we agree that integrating energy-based principles directly into RM training or modifying RL algorithm to directly utilize the uncertainty could offer stronger performance and better alignment. We view this as an exciting direction for future work.
> >
> > REFERENCES
> >
> > [1] Uncertainty-aware Reward Model: Teaching Reward Models to Know What is Unknown, arXiv preprint, 2024
> >
> > [2] Skywork-Reward: Bag of Tricks for Reward Modeling in LLMs, CoRR, 2024

---

> > > ### Author Response · Authors · 2025-06-08
> > > **Gentle Reminder**
> > >
> > > Dear Reviewer,
> > >
> > > Thank you once again for your valuable feedback and questions. We have provided detailed responses and additional empirical results, and we hope these address your concerns.
> > >
> > > As the discussion period concludes, we kindly encourage you to review our responses and **let us know if you have any remaining questions or concerns**.
> > >
> > > We sincerely appreciate your time and effort in helping us improve our work.
> > >
> > > Sincerely,
> > >
> > > Authors

---

> > > > ### Author Response · Authors · 2025-06-11
> > > > **A Gentle Reminder**
> > > >
> > > > Dear Reviewer,
> > > >
> > > > As the discussion period comes to an end, we hope our responses and additional results have addressed your concerns. We kindly encourage you to **review our response** and let us know if any questions remain.
> > > >
> > > > If you have any final thoughts or feedback, we would be grateful to hear them.
> > > >
> > > > Thank you again for your time and effort in helping us improve our work.
> > > >
> > > > Sincerely,
> > > >
> > > > Authors

---

### Official Review · Reviewer_rdh4 · 2025-05-23

**Rating:** 6
**Confidence:** 3
**Ethics Flag:** 1

**Summary:**

This paper introduces Energy-Based Reward Models (EBRM) as a lightweight, post-hoc method to enhance reward models (RMs) used in Reinforcement Learning from Human Feedback (RLHF). Instead of predicting a fixed scalar score, EBRM models the full conditional reward distribution $p(r \mid e)$ using an energy-based formulation. Built on top of a pretrained RM without requiring retraining, EBRM refines reward estimates by leveraging the RM’s embeddings and applying contrastive learning with noise-aware regularization. The framework includes conflict-aware data filtering, hybrid initialization, and iterative inference for reward refinement. Empirical evaluations on RewardBench and RMB benchmarks demonstrate that EBRM consistently improves alignment performance—particularly in safety and reasoning tasks—while mitigating reward hacking during PPO training. The method is modular, scalable, and requires minimal overhead, making it a practical contribution to LLM alignment research.

**Questions To Authors:**

- Why is a fixed initialization range of $$[-2.0, 2.0]$$ chosen, and could adaptive initialization improve generalization?
- How significant is the computational overhead introduced by iterative reward refinement in real-world RLHF settings?
- Does filtering out 25% of misaligned pairs discard valuable training signals, and how do filtered vs. unfiltered variants compare?

**Reasons To Accept:**

- This paper introduces a novel and practical use of conditional energy-based models (EBMs) to refine reward models in RLHF. Unlike prior methods, it improves a pretrained scalar reward model without retraining or ensembles. By modeling the conditional distribution $p(r \mid e) = \frac{\exp(f_\theta(e, r))}{Z_\theta(e)}$, it captures reward uncertainty in a principled way. The method is both elegant and readily integrable into existing systems, combining originality with strong practical value.
- The proposed EBRM method demonstrates strong and consistent empirical performance across various alignment tasks, outperforming both standard and ensemble-based reward models. Notable improvements, such as a 5.97% gain in harmlessness accuracy, highlight its robustness and real-world applicability.
- The technical exposition is clear and well-structured. The derivation of the EBRM objective via noise-aware contrastive estimation (NCE+) is sound, and the motivations behind hybrid initialization and data filtering are well justified. Extensive ablations further enhance the paper’s clarity and reproducibility.
- This work addresses a key challenge in alignment—improving reward models without costly retraining. Its modular, interpretable design, along with demonstrated scalability, positions EBRM as a valuable contribution to both academic research and real-world RLHF systems.

**Reasons To Reject:**

- While the empirical gains are promising, the paper lacks a strong theoretical explanation for why modeling $p(r \mid e)$ with an energy function leads to improved robustness. A deeper comparison with alternatives like quantile regression or DPO would clarify the unique benefits and mechanisms of EBRM.
- The method depends on base RM scores as pseudo-labels, which may introduce bias or noise. Though noise-aware strategies are applied, the absence of external validation (e.g., human labels) makes it hard to judge whether EBRM is correcting or reinforcing RM limitations. Incorporating external calibration would enhance credibility.
- The evaluation focuses on relatively small models, leaving scalability to large, real-world systems untested. Without results on frontier-scale models or broader generalization settings, claims about practical impact remain tentative.
- Robustness to adversarial or ambiguous prompts is not evaluated, despite being a critical alignment concern. Including tests under such conditions would better substantiate the method’s reliability in real-world deployment scenarios.

---

> ### Author Response · Authors · 2025-06-03
> **Response to Reviewer rdh4 (1/3)**
>
> Thank you for the detailed and encouraging review. We are glad you found the method principled, practical, and well-motivated for real-world RLHF applications.
>
> **1. Theoretical Explanation and comparison with quantile regression and DPO**
>
> **1.1 The paper lacks a strong theoretical explanation for why modeling with an energy function leads to improved robustness.**
>
> EBRM improves robustness as it replaces pointwise reward estimation with distributional modeling, enabling it to capture uncertainty in the reward signal.
>
> - **Point estimate vs. distribution modeling:** Standard scalar reward models lack the ability to represent uncertainty, which is critical as human preference data is often noisy, inconsistent, or ambiguous. This limitation leads to overfitting, poor generalization, and susceptibility to reward hacking.
>
> - **Energy-based modeling enables uncertainty:** EBRM defines a conditional reward distribution  $p(r|e) \propto \exp(f_\theta(e,r))$. This allows the model to assign probability mass over a range of plausible rewards, rather than collapsing to a single value. As a result, it captures uncertainty.
>
> - **Uncertainty modeling improves robustness:** EBRM is trained using NCE+ with soft positive labels and negative samples. This training objective encourages the model to learn well-calibrated reward regions, not just match noisy targets. At inference, EBRM refines the Base RM outputs via energy-guided optimization, pushing predictions toward high-likelihood regions of the learned distribution. This mitigates the impact of miscalibrated spurious Base RM scores and improves robustness.
>
> Thus, the robustness of EBRM arises from its ability to model and leverage uncertainty, enabled by the energy-based formulation that showcases the theoretical basis of our approach.
>
>
> **1.2 A deeper comparison with alternatives like quantile regression or DPO would clarify the unique benefits and mechanisms of EBRM.**
>
> Quantile regression [6] models uncertainty by predicting rewards at fixed quantile levels. While this captures distributional information, it learns a discrete set of targets and lacks a continuous density function over rewards. At test time it relies on a weighted average of quantile outputs to approximate the final score.
>
> In contrast, EBRM learns a continuous conditional reward distribution via an energy function, enabling flexible uncertainty modeling and calibrated reward refinement at test time via gradient optimization. This allows EBRM to adjust predictions based on the local reward structure. We do not compare directly to quantile regression methods as the code is not publicly available.
>
> Direct Preference Optimization (DPO), by design, lacks an explicit reward model and assumes scalar scores without modeling uncertainty. EBRM, in contrast, is a post-hoc refinement layer that improves the robustness of existing RMs by modeling uncertainty in scalar reward estimation.
>
> **2. Addressing Pseudo-Label Bias and External Validation**
>
> EBRM does not reinforce RM errors, as demonstrated by consistent gains on human-curated test sets. While it uses the Base RM outputs during training, it mitigates pseudo-label bias through two mechanisms:
>
> i. **Noise-aware training:** Rather than regressing to scalar RM scores, EBRM learns a distribution over plausible rewards by injecting Gaussian noise and contrasting against negative samples. This forces the model to learn a smoother reward landscape based on relative plausibility, not the Base RM's raw score.
> As seen in Figure 5, tuning the noise scale affects performance across tasks: lower noise hurts ambiguous tasks (Chat), while higher noise degrades deterministic ones (Safety). This confirms that EBRM learns to generalize by modeling uncertainty, not memorizing RM bias.
>
> ii. **Filtering corrupted supervision:**
> While EBRM does not rely on external supervision during training, we do leverage existing human preference labels for the training data to filter corrupted supervision. We discard preference pairs where the Base RM assigns a higher reward to the human-rejected response. These cases indicate unreliable or misleading supervision either due to annotation noise or RM bias and are excluded to prevent reinforcement of the Base RM errors.
>
> Further, we validate EBRM externally at test time using two test sets - RewardBench and RMB. The performance gain (up to $\sim$ 6\%) demonstrate that EBRM does not reinforce the Base RM limitations, but actively corrects them through distributional modeling and inference-time reward refinement.

---

> > ### Author Response · Authors · 2025-06-03
> > **Response to Reviewer rdh4 (2/3)**
> >
> > **3. Scalability of EBRM**
> >
> > We agree that benchmarking against stronger RMs is important and have addressed this by extending our experiments to include larger and competitive models.
> >
> > In particular, we now evaluate on **2.6B Pythia Reward Model** and **Skywork-LLaMA3.1-8B** RM [2]. The Skywork RM is ranked \#11 on the RewardBench leaderboard and the strongest publicly available 8B reward model. Results are summarized below:
> >
> > Win rates on RewardBench for the 2.6B Pythia Reward Model.
> > |         | Chat      | Chat-Hard | Safety    | Reasoning | Average   |
> > |---------|-----------|-----------|-----------|-----------|-----------|
> > | Base RM | **81.01** | 35.20     | 38.26     | 73.60     | 57.02     |
> > | EBRM    | 79.61     | **36.73** | **42.69** | **73.84** | **58.22** |
> >
> > Win rates on Reward Model Benchmark (RMB) for the 2.6B Pythia Reward Model.
> > | Method  |           | Harmlessnes |           | Helpfulness | Average   |
> > |---------|-----------|-------------|-----------|-------------|-----------|
> > |         | Pairwise  | BoN         | Pairwise  | BoN         |           |
> > | Base RM | 51.67     | 36.10       | 45.08     | 27.65       | 40.13     |
> > | EBRM    | **53.45** | **36.95**   | **48.89** | **31.23**   | **42.63** |
> >
> > Win rates on RewardBench for the 8B Skywork Reward Model.
> > | Method  | Chat      | Chat-Hard | Safety    | Reasoning | Average   |
> > |---------|-----------|-----------|-----------|-----------|-----------|
> > | Base RM | **94.69** | 88.60     | 92.58     | **96.70** | 93.14     |
> > | EBRM    | **94.69** | **89.69** | **92.66** | **96.70** | **93.43** |
> >
> > Win rates on Reward Model Benchmark (RMB) for the 8B Skywork Reward Model.
> > | Method  |           | Harmlessnes |           | Helpfulness | Average   |
> > |---------|-----------|-------------|-----------|-------------|-----------|
> > |         | Pairwise  | BoN         | Pairwise  | BoN         |           |
> > | Base RM | 72.39     | **56.37**   | 76.36     | 61.64       | 66.69     |
> > | EBRM    | **72.43** | 56.26       | **77.01** | **61.75**   | **66.86** |
> >
> > We observe a consistent positive gain in performance as the Base RM size increases. On the 2.6B Pythia-based RM, EBRM improves average scores on both RMB (+2.5\%) and RewardBench (+1.2\%) tasks. EBRM continues to provide measurable improvement even at the high end of the performance spectrum, delivering +0.29\% on RewardBench and +0.17\% on RMB over the already strong Skywork RM.
> >
> > For comparison, the recent uncertainty-aware method (URM) [1] required full RM retraining, shifting from Bradley-Terry to multi-attribute regression loss and achieved a +0.4\% gain on RewardBench. EBRM, by contrast, is plug-and-play: no retraining, no architecture changes, and still comparable gains.
> > We also note that our Skywork evaluation uses the **cleaned data version**, unlike prior work that unknowingly used a version with RewardBench data leakage [2]. Due to the lack of open-source code, we are unable to reproduce a direct comparison with URM. We also omit ensemble baselines here due to compute constraints.
> >
> > **In summary**, our new results show that EBRM adds value even to top-tier RMs, reinforcing its relevance for practical and scalable alignment refinement.
> >
> > **4. Evaluation on Adversarial or Ambiguous Prompts**
> >
> > We evaluate EBRM on RewardBench and RMB, two benchmarks specifically curated to assess reward model robustness. RewardBench combines different subsets of prompts, including adversarial tasks (under chat-hard). RMB complements this by spanning 49 real-world behavior categories, including contextual, bias, refusal, and more, with both pairwise and Best-of-N evaluations designed to test fine-grained reward distinctions. Prior work shows [3] RMB performance positively correlates with adversarial alignment benchmarks like Arena-Hard and AdvBench.
> >
> > Notably, prior work [1,6,7] evaluated only on RewardBench; we extend this by incorporating RMB to strengthen the empirical rigor of our evaluation. We believe these two datasets provide a comprehensive and challenging testbed for validating EBRM’s generalization and reliability.
> >
> > **5. Why is a fixed initialization range of chosen, and could adaptive initialization improve generalization?**
> >
> > We choose a fixed initialization range $[-2, 2]$ based on the empirical reward distribution of the Base RM, as detailed in Section 4.4 and Table 10. This range covers the vast majority of observed scores and leverages the Base RM’s prior calibration to provide a stable and well-aligned starting point for optimization.
> >
> > If the Base RM’s score falls within this range, we use it directly; otherwise, we sample uniformly from the range to preserve exploration. This design offers a balance between exploiting prior knowledge and ensuring flexibility during refinement.
> >
> > While adaptive initialization is a promising direction for future work, it may introduce additional complexity.

---

> > > ### Author Response · Authors · 2025-06-03
> > > **Response to Reviewer rdh4 (3/3)**
> > >
> > > **6. How significant is the computational overhead introduced by iterative reward refinement in real-world RLHF settings?**
> > >
> > > Following the experimental set up for RL experiment average time for 1 training epoch using the Base RM and EBRM are:
> > > | Method  | Average Time   |
> > > |---------|----------------|
> > > | Base RM | 265.60 seconds |
> > > | EBRM    | 280.07 seconds |
> > >
> > > This shows that EBRM introduces only a $\sim$ 5.4\% increase in per-epoch training time demonstrating that it imposes minimal overhead in RLHF pipelines while offering higher absolute gold scores and stability (Figure 3).
> > >
> > > **7. Does filtering out 25\% of misaligned pairs discard valuable training signals, and how do filtered vs. unfiltered variants compare?**
> > >
> > > Filtering removes training pairs where the Base RM ranks the rejected response higher, these examples reflect conflicted or corrupted supervision, either due to annotation noise or RM miscalibration. Including them can mislead learning and reinforce RM biases.
> > >
> > > **Why these pairs are not learnable:** We cannot determine whether the Base RM misranked the responses, the annotation is flawed, or both. Even if a high quality external RM is used to correctly score the pairs, its score distribution would differ from the Base RM, making the signal inconsistent and uncalibrated.
> > >
> > > **Empirical comparison:** As shown in Figure 6, EBRM trained on the unfiltered dataset still improves over the Base RM, but the filtered version achieves stronger alignment and more stable PPO fine-tuning, confirming that removing these corrupted pairs improves learning.
> > >
> > > **No loss of valuable signal:** In the case of the 1.3B RM, **only 8 examples were filtered**, highlighting that filtering removes conflicted and corrupted supervision.
> > >
> > > Thus, filtering does not discard useful training signals; it removes structurally unreliable supervision, improving generalization and robustness.
> > >
> > > REFERENCES
> > >
> > > [1] Uncertainty-aware Reward Model: Teaching Reward Models to Know What is Unknown, arXiv preprint, 2024
> > >
> > > [2] Skywork-Reward: Bag of Tricks for Reward Modeling in LLMs, CoRR, 2024
> > >
> > > [3] RMB: Comprehensively Benchmarking Reward Models in LLM Alignment
> > >
> > > [4] Two Minds Better Than One: Collaborative Reward Modeling for LLM Alignment, arXiv preprint, 2025
> > >
> > > [5] WorldPM: Scaling Human Preference Modeling, arXiv preprint, 2025
> > >
> > > [6] Quantile Regression for Distributional Reward Models in RLHF, arXiv preprint, 2024
> > >
> > > [7] RRM: Robust Reward Model Training Mitigates Reward Hacking, arXiv, 2024

---

> > > > ### Author Response · Authors · 2025-06-08
> > > > **Gentle Reminder**
> > > >
> > > > Dear Reviewer,
> > > >
> > > > Thank you once again for your valuable feedback and questions. We have provided detailed responses and additional empirical results, and we hope these address your concerns.
> > > >
> > > > As the discussion period concludes, we kindly encourage you to review our responses and **let us know if you have any remaining questions or concerns**.
> > > >
> > > > We sincerely appreciate your time and effort in helping us improve our work.
> > > >
> > > > Sincerely,
> > > >
> > > > Authors

---

> > > > > ### Author Response · Authors · 2025-06-11
> > > > > **A Gentle Reminder**
> > > > >
> > > > > Dear Reviewer,
> > > > >
> > > > > As the discussion period comes to an end, we hope our responses and additional results have addressed your concerns. We kindly encourage you to **review our response** and let us know if any questions remain.
> > > > >
> > > > > If you have any final thoughts or feedback, we would be grateful to hear them.
> > > > >
> > > > > Thank you again for your time and effort in helping us improve our work.
> > > > >
> > > > > Sincerely,
> > > > >
> > > > > Authors

---

### Official Review · Reviewer_JZMs · 2025-05-23

**Rating:** 6
**Confidence:** 3
**Ethics Flag:** 1

**Summary:**

This paper introduces an energy-based reward model (EBRM), a lightweight, post-hoc framework designed to enhance robustness and generalization of reward models in the context of RLHF. EBRM is built on top of a base reward model by extracting the embeddings $e$ at the penultimate layer of this RM and using it to model the conditional probability of rewards given these embeddings $f_{\theta}(e,r)$. This contrasts with the usual reliance on single scalar rewards, which authors claim enables EBRM to capture a richer reward landscape and identify implausible reward assignments for the base reward model.
To train EBRM, the authors employ three techniques: (1) conflict-aware data filtering that removes ~25% of preference pairs where the base RM contradicts human annotations, (2) label-noise-aware contrastive training using NCE+ that treats base RM rewards as noisy rather than exact, and (3) negative sampling from Gaussian distributions to help the model distinguish between plausible and implausible rewards.
During inference, EBRM refines rewards through energy-guided optimization: given a prompt-response pair, the base RM produces an initial reward and embedding, then EBRM uses gradient ascent (up to 50 steps) to find the reward value that maximizes the learned energy function $f_{\theta}(e,r)$. The method includes hybrid initialization that uses the base RM's reward when within a given range, otherwise samples from a uniform distribution.
The authors evaluate EBRM on reward modeling benchmarks (RewardBench, RMB) and downstream RL experiments using 44M and 1.3B parameter models. Results show consistent but modest improvements of 1-6% across tasks, with particularly notable gains in safety-critical scenarios (up to 5.97% on harmlessness for the smaller model). In PPO experiments, EBRM delays reward hacking and achieves higher gold reward scores compared to base RMs and ensemble methods. The energy-based component adds ~3% to the base RM's parameter count, though inference requires additional computational overhead for the optimization procedure.

**Questions To Authors:**

- *Scaling Limitation* How do you expect results to transfer to larger models (7B+) given testing only inconclusively up to 1.3B parameters? Can you provide arguments for why EBRM would remain effective when embedding quality and reward calibration change at lerger scales?
- *Baselines and Statistical Significance* How does EBRM compare to other uncertainty-aware methods and are your improvements statistically significant?
- *Circular Dependency* How can EBRM correct systematic biases when it learns to trust/distrust the base RM using the base RM's own outputs and embeddings? Have you tested on base RMs with known systematic biases, and when would you expect the method to help vs. hurt?
- *Next Steps* What evidence would convince users to adopt EBRM over simpler alternatives like better training or ensembles? What are the 2-3 most important experiments you would prioritize to address the scale limitations and improve this method?

**Reasons To Accept:**

- *Novel, Principled Approach*: The paper addresses reward overoptimization, a fundamental RLHF challenge that will grow more important as models scale. The application of energy-based models to reward modeling is genuinely innovative, representing a theoretically grounded shift from scalar to distributional reward modeling. The post-hoc design is particularly valuable, enabling immediate deployment to existing systems without costly retraining.
- *Consistent Empirical Results*: The method shows modest but consistent improvements (1-6%) across multiple benchmarks, with particularly strong performance on safety tasks. Additionelly, PPO experiments demonstrate that EBRM delays reward hacking and achieves higher gold reward scores - a practically important result for RLHF stability.
- *Technical Execution*: The authors demonstrate thoughtful design with well-motivated choices including NCE+ for handling noisy labels, hybrid initialization, and conflict-aware filtering. The lightweight architecture (3% parameter overhead) makes adoption feasible, and the comprehensive evaluation includes both reward model benchmarks and downstream RL validation.
- *Transparency and Future Research Directions*: The paper provides honest reporting of limitations, sufficient technical detail for reproducibility, and works within realistic computational constraints. This opens a new research direction at the intersection of energy models and alignment, providing a framework for future uncertainty-aware reward modeling work.
- *Writing Quality*: The paper is very well written and extremely clear.

**Reasons To Reject:**

- *Cost-Benefit Trade-off*: 1-6% gains may not justify adding 50+ optimization steps per inference. The performance actually drops on some tasks (chat) and seems not to be consistent when scaling the model size, suggesting fundamental limitations that undermine claims of general robustness enhancement.
- *Methodology*: The aggressive data filtering removes 25% of cases where the base RM disagrees with human preferences - precisely the challenging scenarios where robustness is most needed. This introduces circular dependency: the method learns when to trust the base RM using the base RM's own outputs and embeddings, potentially limiting its ability to correc biases.
- *Limited Generalizability*: Testing only up to 1.3B parameters limits practical relevance, when production systems use much larger models. There's no evidence the approach scales appropriately, and embedding quality and reward calibration may behave differently at scale. Moreover, the experiment on 1.3B model seems to be less conclusive than the smaller one
- *Baselines*: The evaluation lacks comparison to other uncertainty-aware RM methods, confidence intervals failure analysis. The authors don't investigate when the method might hurt performance or fail to help.

---

> ### Author Response · Authors · 2025-06-03
> **Response to Reviewer JZMs (1/3)**
>
> Thank you for the comprehensive and thoughtful review. We appreciate your recognition of the method’s novelty, clarity, and practical relevance.
>
> **1. Cost-Benefit Trade-off**
>
> We believe EBRM offers a compelling cost-benefit trade-off, especially compared to alternatives like ensembling and retraining.
>
> i. EBRM increases inference latency by only $\sim$ 2× compared to the Base RM. In contrast, ensembles typically incur $\sim$ n× the inference time and parameter count, where n $\ge$ 3 in practice. Retraining full RMs is substantially more expensive in both compute and time. EBRM can be trained in just 465 seconds from a 70M Pythia model with only 3\% the size of a standard RM.
>
> ii. Across five alignment tasks, EBRM **outperforms ensembles** by 1–5\% while being 34\% faster and adding $<$3\% in parameter count. These gains are non-trivial given that ensembles are a strong and widely adopted baseline.
>
> iii. EBRM requires no retraining and is applied post hoc to any RM, making it a lightweight and practical solution for deployment scenarios where retraining or maintaining ensembles is infeasible.
>
> **2. Drop in Chat Accuracy**
>
> The dip is not unique to EBRM. Similar declines appear in both uncertainty-aware [1,3] and standard methods [4]. **This suggests that the issue arises from task ambiguity, not EBRM design**. Open-ended tasks are inherently ambiguous, similar inputs can yield diverse valid preferences. EBRM’s noise-aware contrastive loss reflects this ambiguity by flattening the energy surface, leading to lower confidence but better calibrated uncertainty. To validate this interpretation, we analyzed the shape of EBRM’s reward distributions for a subset of each RewardBench task:
>
> i. Kurtosis indicates a distribution's peakedness: $3$ indicates normal distribution, $>3$ (leptokurtic) implies sharper, more confident predictions, and $<3$ (platykurtic) indicates flatter, more uncertain ones.
>
> ii. Variance captures the distribution’s spread.
>
>
> Mean and standard deviation of the variance and kurtosis values.
> | Task      | Mean Var. | Std. Var. | Mean Kurtosis | Std. Kurtosis | Kurtosis Type |
> |-----------|-----------|-----------|---------------|---------------|---------------|
> | Chat      | 3.14      | 1.33      | 2.56          | 0.56          | Platykurtic   |
> | Chat-Hard | 1.18      | 0.54      | 3.90          | 0.65          | Leptokurtic   |
> | Safety    | 1.86      | 0.82      | 3.15          | 0.41          | Mesokurtic    |
> | Reasoning | 1.58      | 0.24      | 3.34          | 0.16          | Leptokurtic   |
>
> Chat exhibits the highest variance and a platykurtic distribution, indicating that EBRM correctly models the ambiguity in these tasks. While this may reduce pairwise win rates, it improves the model’s calibration and mitigates the risk of overfitting.
>
>
> **3.  Methodology: Circular Dependency and Filtering**
>
> Our data filtering strategy is not circular nor aggressive—it improves training signal quality by removing contradictory or corrupted supervision, not difficult cases.
>
> - The filtered examples were excluded because they provide misleading signals that degrade EBRM's performance. As shown in Figure 6, without filtering, the performance of EBRM will slightly degrade. While it is possible to correct such examples, we are unaware of a principled way to do so, as it is often unclear whether the error lies with the Base RM, the human annotation, or both. Additionally, using high-quality external RMs for correction could introduce reward scale mismatches, leading to further inconsistency in EBRM’s training data.
>
> - The 25\% filtering only applies to the small RM (70M), where the capacity is not high. For the 1.3B RM, only 8 pairs were filtered. This clearly shows that filtering is not “aggressive”, but a step to avoid injecting harmful signals into training.
>
> - EBRM does not blindly trust the Base RM. In fact, it treats the Base RM’s outputs as noisy priors and uses noise-aware training to learn the distribution rather than regressing to the Base RM. This allows EBRM to correct biases instead of reinforcing them.
>
> In summary, filtering improves EBRM by training on high-quality, consistent signals. Combined with noise-aware training, it enables EBRM to identify and correct the Base RM errors instead of reproducing them.

---

> > ### Author Response · Authors · 2025-06-03
> > **Response to Reviewer JZMs (2/3)**
> >
> > **4. Limited Generalizability**
> >
> > We agree that benchmarking against stronger RMs is important and have addressed this by extending our experiments to include larger and competitive models.
> >
> > We now evaluate on **2.6B Pythia Reward Model** and **Skywork-LLaMA3.1-8B** RM [2]. The Skywork RM is ranked \#11 on the RewardBench leaderboard and the strongest publicly available 8B reward model. Results are summarized below:
> >
> > Win rates on RewardBench for the 2.6B Pythia Reward Model.
> > |         | Chat      | Chat-Hard | Safety    | Reasoning | Average   |
> > |---------|-----------|-----------|-----------|-----------|-----------|
> > | Base RM | **81.01** | 35.20     | 38.26     | 73.60     | 57.02     |
> > | EBRM    | 79.61     | **36.73** | **42.69** | **73.84** | **58.22** |
> >
> > Win rates on Reward Model Benchmark (RMB) for the 2.6B Pythia Reward Model.
> > | Method  |           | Harmlessnes |           | Helpfulness | Average   |
> > |---------|-----------|-------------|-----------|-------------|-----------|
> > |         | Pairwise  | BoN         | Pairwise  | BoN         |           |
> > | Base RM | 51.67     | 36.10       | 45.08     | 27.65       | 40.13     |
> > | EBRM    | **53.45** | **36.95**   | **48.89** | **31.23**   | **42.63** |
> >
> > Win rates on RewardBench for the 8B Skywork Reward Model.
> > | Method  | Chat      | Chat-Hard | Safety    | Reasoning | Average   |
> > |---------|-----------|-----------|-----------|-----------|-----------|
> > | Base RM | **94.69** | 88.60     | 92.58     | **96.70** | 93.14     |
> > | EBRM    | **94.69** | **89.69** | **92.66** | **96.70** | **93.43** |
> >
> > Win rates on Reward Model Benchmark (RMB) for the 8B Skywork Reward Model.
> > | Method  |           | Harmlessnes |           | Helpfulness | Average   |
> > |---------|-----------|-------------|-----------|-------------|-----------|
> > |         | Pairwise  | BoN         | Pairwise  | BoN         |           |
> > | Base RM | 72.39     | **56.37**   | 76.36     | 61.64       | 66.69     |
> > | EBRM    | **72.43** | 56.26       | **77.01** | **61.75**   | **66.86** |
> >
> > We observe a consistent positive gain in performance as the Base RM size increases. On the 2.6B Pythia-based RM, EBRM improves average scores on both RMB (+2.5\%) and RewardBench (+1.2\%) tasks. EBRM continues to provide measurable improvement even at the high end of the performance spectrum, delivering +0.29\% on RewardBench and +0.17\% on RMB over the already strong Skywork RM.
> >
> > For comparison, the recent uncertainty-aware method (URM) [1] required full RM retraining, shifting from Bradley-Terry to multi-attribute regression loss and achieved a +0.4\% gain on RewardBench. EBRM, by contrast, is plug-and-play: no retraining, no architecture changes, and still comparable gains.
> > We also note that our Skywork evaluation uses the **cleaned data version**, unlike prior work that unknowingly used a version with RewardBench data leakage [2]. Due to the lack of open-source code, we are unable to reproduce a direct comparison with URM. We also omit ensemble baselines here due to compute constraints.
> >
> > **In summary**, our new results show that EBRM adds value even to top-tier RMs, reinforcing its relevance for practical and scalable alignment refinement.
> > Further, per-task gains vary slightly with scale due to differences in task ambiguity, difficulty, and the Base RM, the overall improvements remain stable and strictly positive across all model sizes.
> >
> > **5. Baseline Comparison**
> >
> > Our baseline choices are grounded in practical constraints and design considerations.
> >
> > - Existing uncertainty-aware RMs do not have public implementations, which prevents direct comparison. Moreover, these methods typically require full retraining of the reward model, which is orthogonal to EBRM’s post-hoc and lightweight design.
> >
> > - Instead, we benchmark against ensembles, which are widely recognized for improving RM's robustness. While ensemble-based approaches are not strictly post-hoc as they require multiple trained reward models, they represent a similar strategy for refining reward signals without modifying the core RM architecture.
> >
> > - Additionally, we include an indirect comparison to an existing uncertainty-aware method [1] via the Skywork-8B RM [2] in Response 4.  EBRM achieves comparable improvements on this strong baseline (+0.29\% on RewardBench, +0.17\% on RMB), demonstrating its effectiveness even without any RM retraining.
> >
> > **6. When does EBRM hurt performance or fail to help?**
> >
> > We observe that EBRM shows slight performance declines on highly ambiguous tasks such as Chat, where multiple responses can be equally valid and human preferences are inconsistent. In such settings, the lack of clear supervision leads EBRM to broaden the energy landscape (as discussed in Response 2), reducing confidence to reflect uncertainty at the cost of pairwise accuracy.
> > Additionally, if the Base RM is severely miscalibrated and fails to provide any meaningful ranking signal, EBRM has limited ability to learn a useful reward landscape from its outputs.

---

> > > ### Author Response · Authors · 2025-06-03
> > > **Response to Reviewer JZMs (3/3)**
> > >
> > > **7. Evaluation on RMs with Systemic Bias**
> > >
> > > Quantifying systemic bias in RMs is nontrivial, so we deliberately use small reward models (e.g., 70M and 1.3B Pythia) in our experiments because their limited capacity makes them more prone to overconfidence and spurious correlations. These small RMs are more likely to rely on shallow heuristics and dataset artifacts to make preference judgments, making them practical for evaluating EBRM’s robustness to bias. EBRM’s consistent improvement over the Base RM and the ensemble variants (known to reduce bias and reliance on spurious correlations) highlights its ability to mitigate bias.
> > >
> > >
> > > **8 Next Steps**
> > >
> > > **8.1 What evidence would convince users to adopt EBRM over simpler alternatives like better training or ensembles?**
> > >
> > > EBRM delivers consistent performance gains across RM sizes from 70M to 8B (see Response 4), while incurring significantly lower cost than retraining or ensemble methods (see Response 1). It adds less than 3\% parameter overhead, requires no RM modification, and improves performance even on strong baselines like Skywork-8B making it a cheaper and generalizable post-hoc alternative over retraining and ensembles.
> > >
> > > **8.2 What are the 2-3 most important experiments you would prioritize to address the scale limitations and improve this method?**
> > >
> > > We have demonstrated EBRM’s scalability with consistent gains across RM sizes from 70M to 8B (see Response 4). As future directions, we would prioritize the following to further strengthen EBRM:
> > >
> > > -  Incorporate EBRM’s uncertainty into RLHF to exploit EBRM’s reward distributions to guide policy updates more robustly.
> > >
> > > - Extend EBRM generalization to handle domain shift by training on available preference data without needing access to the Base RM’s training set.
> > >
> > > REFERENCES
> > >
> > > [1] Uncertainty-aware Reward Model: Teaching Reward Models to Know What is Unknown, arXiv preprint, 2024
> > >
> > > [2] Skywork-Reward: Bag of Tricks for Reward Modeling in LLMs, CoRR, 2024
> > >
> > > [3] Quantile Regression for Distributional Reward Models in RLHF, arXiv preprint, 2024
> > >
> > > [4] RRM: Robust Reward Model Training Mitigates Reward Hacking, arXiv, 2024
> > >
> > > [5] Two Minds Better Than One: Collaborative Reward Modeling for LLM Alignment, arXiv preprint, 2025
> > >
> > > [6] WorldPM: Scaling Human Preference Modeling, arXiv preprint, 2025

---

> > > > ### Comment · Reviewer_JZMs · 2025-06-04
> > > >
> > > > Thank you for your detailed response. In light of the discussion and new results, I raise my score to 6.

---

### Author Response · Authors · 2025-06-03
**Official Comment by Authors**

We thank the reviewers for their detailed feedback. Below, we summarize the main questions raised and how they have been addressed in the rebuttal:

- We added new experimental results on **2.6B Pythia RM** and **Skywork-LLaMA3.1-8B**, demonstrating that EBRM scales effectively to stronger base RMs and consistently improves accuracy, even on high-performing models.

- We clarified the rationale behind data filtering and explained how EBRM mitigates pseudo-label bias through noise-aware training.

- We provided justification for our choice of baselines and evaluation datasets.

- We elaborated on the cost-benefit trade-off of EBRM.

We would like to highlight the novelty and contributions of our work. Ensuring robust and reliable reward modeling is critical for safe and reliable alignment in RLHF, yet existing reward models often suffer from overconfidence and poor generalization. Our work presents a lightweight, post-hoc solution.

- We identify that a key limitation of standard scalar reward models is their reliance on point estimates, which cannot express uncertainty or resolve conflicting supervision.

- We address this by modeling a reward distribution via an energy-based function, allowing the model to capture uncertainty and calibrate its reward signal accordingly.

- We further utilize two techniques to improve reward robustness: (1) conflict-aware filtering that removes supervision where the Base RM contradicts human preferences, and (2) noise-aware contrastive training (NCE+).

- Our method is plug-and-play and model agnostic.

- We demonstrate that EBRM **scales effectively** and **improves performance** across diverse tasks and RL experiments, and offers **better cost-benefit tradeoffs** than ensembles or retraining-based approaches.

We hope these contributions, combined with a simple, plug-and-play design and significant empirical improvements, offer a valuable step forward for trustworthy reward modeling. We appreciate the reviewers’ feedback and would be grateful for any further insights or suggestions.

---

### Decision · Program_Chairs · 2025-07-08

**Decision:**

Accept

**Comment:**

The authors introduce a lightweight framework inspired by energy based models to improve reward model robustness and generalization post-hoc. They demonstrate the effectiveness of this across several reward modeling benchmarks and model scales. Iniital concerns raised by the reviewers on the comprehensiveness of evaluation were adequately addressed during the rebuttal period. Hence I recommend acceptance.